# An Optimum Design of Clocked AC-DC Charge Pump Circuits for Vibration Energy Harvesting

**Jinming Ye [†] and Toru Tanzawa ***

Faculty of Engineering, Shizuoka University, Hamamatsu 432-8561, Japan; zxc324691@gmail.com
* Correspondence: toru.tanzawa@shizuoka.ac.jp
† Current address: Graduate School of Biomedical Engineering, Tohoku University, Sendai 980-8579, Japan.

**Abstract:** This paper shows how clocked AC-DC charge pump circuits can be optimally designed to have the minimum circuit area for small form factor vibration energy harvesting. One can determine an optimum number of stages with simple equations and then determine the capacitance of each pump capacitor to have a target output current at a target output voltage. The equations were verified under a wide range of design parameters by comparing the output current with the simulated one. The output current of the circuit designed by the equations was in good agreement with the simulated result, to within 5% for 98% of the 1600 designs with different parameters. We also propose a design flow to help designers determine the initial design parameters of a clocked AC-DC charge pump circuit (i.e., the number of stages, capacitance per stage, and the total size of rectifying devices) under the condition that the saturation current of a unit of the rectifying device, clock frequency, amplitude of the voltage generated by the energy transducer, target output voltage, and target output current are given. SPICE simulation results validated theoretical results with an error of 3% in terms of the output current when a clocked AC-DC charge pump was designed to output current of 1 µA at 2.5 V from a vibration energy harvester with an AC voltage amplitude of 0.5 V.

**Keywords:** vibration energy harvesting; clocked AC-DC charge pump; optimum design; design flow

## 1. Introduction

Energy harvesting (EH) is a technique used to harvest electrical energy from other types of ambient energy such as light, heat flow, electromagnetic waves, and kinetic energy [1,2]. When an energy transducer (ET), such as a photovoltaic or thermoelectric generator, generates DC power, a sensor integrated circuit (IC) needs a DC-DC power conversion. When an ET, such as a vibration power generator, generates AC power, the sensor IC needs an AC–DC power conversion. A vibration ET (VET) utilizes various types of material, including piezoelectric [3], electrostatic [4], and magnetostrictive [5] materials. Among them, magnetostrictive VET has the highest reliability because of the use of robust materials such as Fe–Ga alloy. A disadvantage of magnetostrictive VETs over electrostatic and piezoelectric VETs is that the open circuit voltage can be low, in the order of 100 mV. Therefore, it requires an AC-DC boost converter [6,7]. To date, various designs have been proposed for VETs [8–11]: AC-DC charge pumps with discrete capacitors and diodes have been used on the printed board [8]; a two-chip solution was proposed and was verified using a VET [9]; an active diode was presented to reduce a voltage drop at the interface [10]; in [11], a switching regulator was designed with a rectifier. AC-DC charge pumps have been used for wireless power transfer via microwaves [12,13]. The integration of AC-DC charge pumps is possible simply because the frequency of the RF power is very high, at over 100 MHz, where the output current from the AC-DC charge pumps is proportional to the frequency and the capacitance of each boosting capacitor. The high frequency enables all of the boosting capacitors for the charge pumps to be integrated. Conversely, AC-DC charge pumps have not been integrated for

VET due to the low AC power frequency, which is lower than 1 kHz. An extremely low frequency requires large capacitors such as chip capacitors. Consequently, clocked AC-DC charge pumps were proposed and developed for magnetostrictive EH [14,15]. In [14], the concept was proposed and a relationship between output voltage and current was presented, in which it was assumed that the AC power supply is ideal with zero impedance. In [15], a circuit system including a clocked AC-DC charge pump and a magnetostrictive ET was developed and experimental results were shown where the model was extended to include the output impedance of VET. Two different definitions for power efficiency were used for comparison: (1) the ratio of DC output power of a clocked AC-DC charge pump ($P_{OUT}{}^{CP}$) to the maximum available output power of VET ($P_{OUT\_MAX}{}^{ET}$), i.e., overall power efficiency $\eta_{SYS} \equiv P_{OUT}{}^{CP}/P_{OUT\_MAX}{}^{ET}$; (2), the ratio of $P_{OUT}{}^{CP}$ to the input power of the charge pump ($P_{IN}{}^{CP}$), i.e., charge pump power efficiency $\eta_{CP} \equiv P_{OUT}{}^{CP}/P_{IN}{}^{CP}$. Many publications have used charge pumps for energy harvesting [16–23] and others have discussed the design methodology for charge pumps [24–30], yet none have discussed the optimal design of clocked AC-DC charge pumps.

In this paper, an extended version of a conference paper [31], we discuss the optimum design and design flow for clocked AC-DC charge pumps in more detail. We present how to determine the optimum number of stages for clocked AC-DC charge pump circuits and the capacitance of each pump capacitor required to reach a target output current and output voltage with the minimum circuit area for small-form-factor vibration EH. The overall design flow is also discussed. In Section 2, we provide design equations to determine the optimum number of stages to obtain the minimum circuit area. In Section 3, the design equations are validated by comparing predictions with SPICE simulations. In Section 4, an entire design flow is outlined, and in Section 5 we discuss the further study required. Without such design optimization, each circuit designer must run many SPICE simulations to determine the circuit parameters to meet a given condition for the input voltage amplitude, output voltage and output current with minimum circuit area. With the proposed novel design optimization, one can determine the circuit parameters required to minimize the circuit area without running many SPICE simulations.

## 2. Optimum Design of Clocked AC-DC Charge Pumps

Figure 1 shows a clocked AC-DC charge pump together with a VET. The full bridge rectifier (FBR) is based on a cross-coupled CMOS bridge circuit [27]. A rectified voltage $V_{REC}$ is multiplied by a digital clock signal in a ring oscillator (ROSC) to generate driving signals *CLK* and *CLKB* for the charge pump (CP). Diode-connected CMOS transistors are used for the diode portion of CP [32]. The oscillator frequency f is controlled by a current source $I_{OSC}$. In [14,15], the relationship between the output voltage ($V_{OUT}$) and current ($I_{OUT}$) is given to be (1) when parasitic capacitance of pumping capacitors is sufficiently small;

$$I_{OUT} = \frac{2fC}{\pi N}[NV_{DD}\sin(\pi/2 - \theta_S) - (\pi/2 - \theta_S)\big((N+1)V_{TH}^{EFF} + V_{OUT}\big)], \tag{1}$$

where $C$, $N$, $V_{DD}$, $\theta_S$ and $V_{TH}^{EFF}$ are the capacitance of each pump capacitor, the number of stages, the amplitude of the open circuit voltage of VET, the conduction angle defined by the point where the output current starts flowing, and the effective threshold voltage, respectively. The point at which output current starts flowing can be calculated by

$$\theta_S = \sin^{-1}\big(\frac{V_{OUT} + (N+1)V_{TH}^{EFF}}{V_{DD}(N+1)}\big), \tag{2}$$

The effective threshold voltage is calculated by

$$V_{TH}^{EFF} = V_T \ln(4^{\frac{1}{N+1}} \frac{fSV_T}{NI_S}). \tag{3}$$

where $V_T$, $S$ and $I_S$ are the thermal voltage, the silicon area given by CN, and the saturation current of each stage diode, respectively.

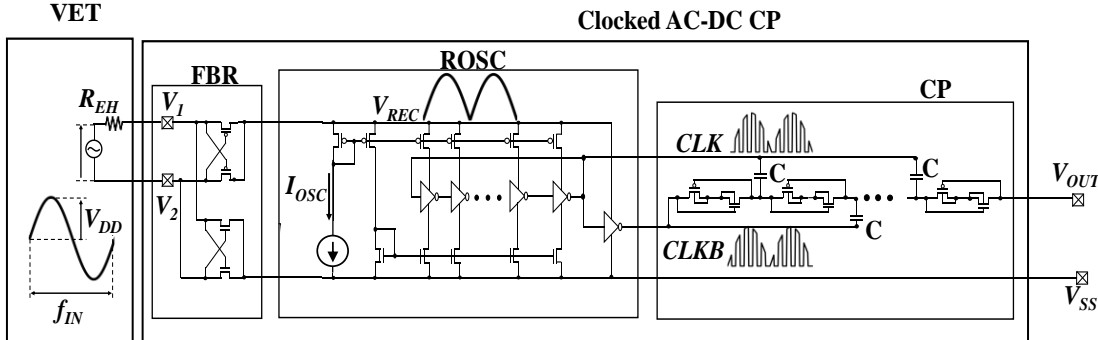

**Figure 1.** Clocked AC-DC charge pump with vibration energy transducer (VET).

Equations (1)–(3) were presented in [14,15]. As the circuit parameters are included in such complicated equations, it is not easy to find an optimum design parameter set of *C*, *N* and diode size at the initial design phase when $V_{DD}$ and a target $I_{OUT}$ at $V_{OUT}$ are given. Therefore, estimation equations should be useful for designers who want to have an initial estimate for those design parameters. In other words, one can determine $N_{OPT}$ by running many SPICE simulations to find the peak point of $I_{OUT}$. It must be beneficial for designers who want to identify the optimum number of stages, $N_{OPT}$, to minimize the circuit area using a single calculation.

As Equations (1)–(3) are extracted based on the formula for DC-DC charge pumps, as shown in [24], we begin by determining the relationship between the minimum number of stages, $N_{MIN}$, required to hold $I_{OUT}$ ($N = N_{MIN}$) = 0, and the optimum number of stages, $N_{OPT}$, required to minimize the circuit area, i.e., $N_{OPT} = 2 N_{MIN}$. Then, assuming that $N_{OPT}$ is proportional to $N_{MIN}$ for clocked AC-DC charge pumps as well as DC-DC charge pumps, $N_{OPT} = k N_{MIN}$, where k is a proportional coefficient determined empirically. As in [15], the following parameters were used to generate the curve in Figure 2a with Equations (1)–(3): $f$ = 10 MHz, $V_{DD}$ = 0.5 V, $V_{OUT}$ = 2.5 V, $V_T$ = 34 mV, $S$ = 10 pF, and $I_S$ = 10 nA. Under a given circuit area of S, $I_{OUT}$ was calculated with Equations (1)–(3). Figure 2a shows that the design with k~2.6 should provide high $I_{OUT}$ with a sufficient design margin when (4) is used for $N_{MIN}$.

$$N_{MIN} = \frac{V_{OUT}}{V_{DD} - V_T \ln(fSV_T/I_S)} - 1. \tag{4}$$

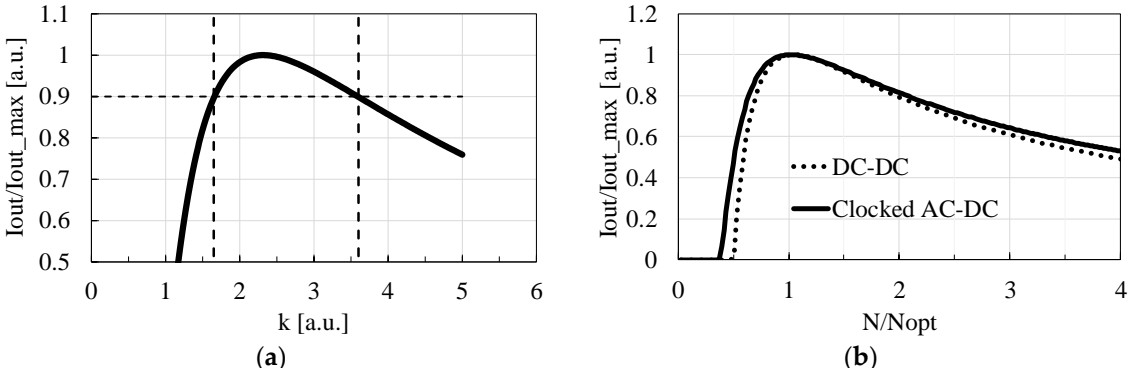

**Figure 2.** *k* vs. $I_{OUT}$ for the clocked AC-DC charge pump (**a**) and $N/N_{OPT}$ vs. $I_{OUT}$ for the clocked AC-DC and DC-DC charge pumps with the same design parameters (**b**).

Even though $k$ varies by 1 in a different parameter set, $I_{OUT}$ only decreases by 10%. Therefore, one can use the optimization Equation (5) together with Equation (4) to determine the number of stages of clocked AC-DC charge pumps, which is the most important product of this paper.

$$N_{OPT} = 2.6 N_{MIN} \tag{5}$$

Figure 2b compares the sensitivity of $I_{OUT}$ on variation in $N$ of the clocked AC-DC and DC-DC charge pumps whose design parameters are common. Note that the clocked AC-DC charge pump is less sensitive to variation in $N$ than the DC-DC charge pump.

## 3. Validation of the Proposed Design

### 3.1. Validation of the Analytical $I_{OUT}$-$V_{OUT}$ Equation

To validate the analytical $I_{OUT}$-$V_{OUT}$ Equation (1) together with Equations (2)–(3), the measured data from the previously fabricated clocked AC-DC charge pump [15] were used. The design parameters were $V_{DD} = 0.5$ V, $N = 9$, $C = 10$ pF, $Is = 40$ nA, $V_T = 34$ mV and $f = 4.8$ MHz. Figure 3 compares the results from model 1 with the measured data. The discrepancy was within 10%, in the range of $V_{DD} = 0.4$–0.8 V and $V_{OUT} = 1$–3.5 V. Model 1 is therefore valid for design optimization based on Equation (1).

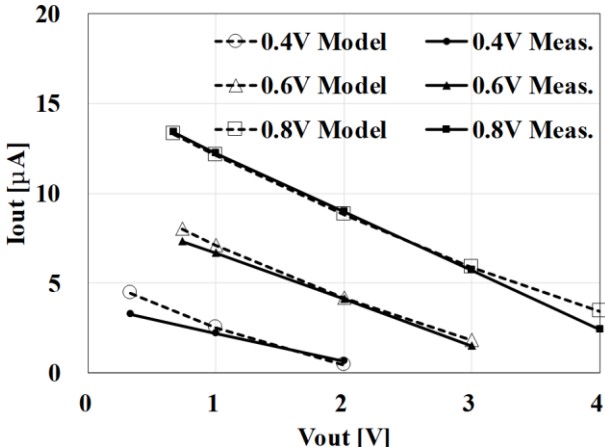

**Figure 3.** Comparison of $V_{OUT}$–$I_{OUT}$ between model Equations (1)–(3) and measured data for a previously fabricated clocked AC-DC charge pump [15].

To further validate the model Equations (1)–(3) at different numbers of stages, SPICE simulations were run under the conditions of $f = 1$ MHz, $V_{DD} = 0.5$ V, $V_{OUT} = 2.0$ V, $V_T = 34$ mV, $S = 1$ nF, and $I_S = 10$ nA. Figure 4 compares $I_{OUT}$ values when $N$ is varied and $S$ is constant. The model results are in good agreement with SPICE simulations. Figure 4 indicates that $N_{OPT}$ is 20, whereas (4) and (5) suggest $N_{OPT}$ is 25. Even though $N_{OPT}$ differed by 20%, the discrepancy in $I_{OUT}$ is as small as 2.5% thanks to the moderate curvature around the highest point in $I_{OUT}$.

### 3.2. Validation of the Optimization Equation

To validate whether the optimization equation (5) together with Equations (4) is effective for designing clocked AC-DC charge pumps, the parameters shown in Table 1 were investigated. $C$ and $N$ were calculated by a numerical simulation to have the highest $I_{OUT}$ under 1600 different combinations of design parameters based on Equations (1)–(3). $C$ and $N$ were also calculated based on Equations (4) and (5) and $S = CN$. Then, the values for $I_{OUT}$ calculated by those methods were compared. The results are shown in Figure 5a, which indicates that model Equations (4) and (5) for 800 of the combinations of parameters were in good agreement, to within 2% of numerical simulations using Equations (1)–(3), and that 98% of all the parameter combinations exhibited differences under 5%. When $V_T \ln(fSV_T/I_S)$

becomes close to $V_{DD}$, the error increases. As this situation occurs for $N_{MIN} > 100$, such cases were omitted from the potential design conditions. Figure 5b shows that the discrepancy in values between models increases as the frequency increases to over 10 MHz. If a designer wishes to run clocked AC-DC charge pumps at high frequencies, Equations (4) and (5) may not be suitable.

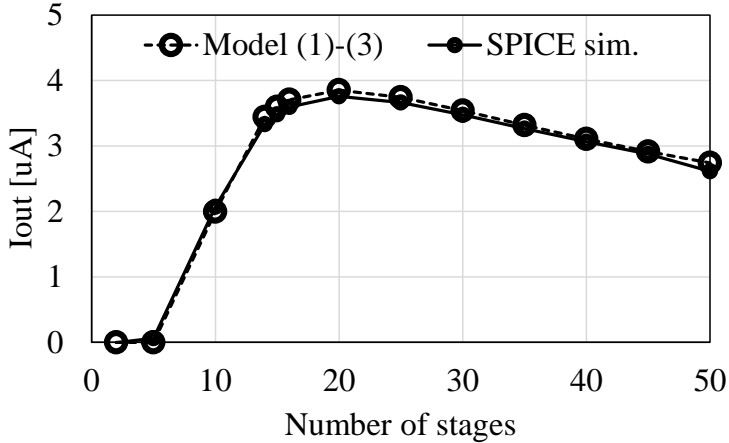

**Figure 4.** Comparison of $I_{OUT}$ vs. $N$ between the models and SPICE simulations.

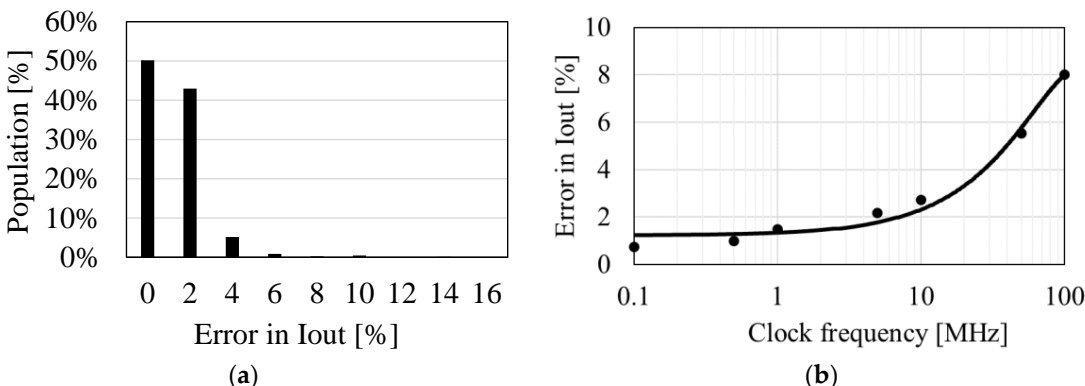

(**a**)            (**b**)

**Figure 5.** Distribution of errors in $I_{OUT}$ between the two methods (**a**) and errors in $I_{OUT}$ vs. clock frequency (**b**).

**Table 1.** Range of parameters used.

| Design Parameters | Maximum | Minimum |
|:---:|:---:|:---:|
| $f$ | 100 MHz | 100 kHz |
| $V_{DD}$ | 0.7 V | 0.5 V |
| $V_{OUT}$ | 3.5 V | 1.5 V |
| $I_S$ | 100 nA | 5 nA |
| $S$ | 1 nF | 1 pF |

## 4. Design Flow of Clocked AC-DC Charge Pumps

Figure 6 proposes a design flow for clocked AC-DC charge pumps. At step 1, the condition is defined where $I_{OUT}{}^{\text{Target}}$ is the target output current at $V_{OUT}$. At step 2, $N_{OPT}$ is calculated by using Equations (4) and (5). At step 3, $C_{OPT}$ value is calculated by S divided by $N_{OPT}$. At step 4, $I_{OUT0}$ is defined by $I_{OUT}$ with $C_{OPT}$ and $N_{OPT}$ via Equations (1)–(3). At step 5, the scaling factor $\lambda$ is identified. At step 6, the final values $I_{SFIN}$, $C_{FIN}$, and $S_{FIN}$ are scaled with $\lambda$. As a result, all the design parameters are determined. Those values are used for the initial SPICE simulation to verify whether the initial estimate is acceptable. Otherwise, the parameters may need to be updated.

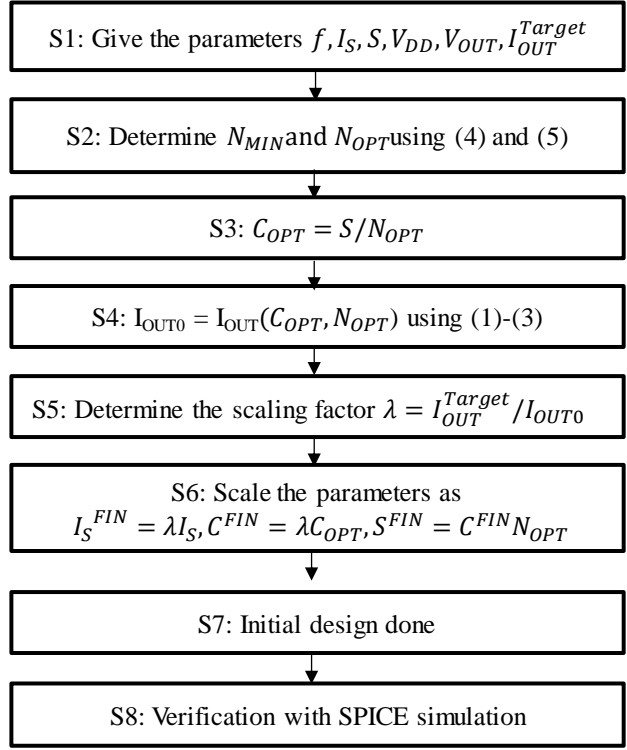

**Figure 6.** Design flow for clocked AC-DC charge pumps.

To provide an example, we demonstrate the above design flow under the conditions of $f$ = 1 MHz, $V_{DD}$ = 0.5 V, $V_{OUT}$ = 2.0 V, $VT$ = 34 mV, $S$ = 1 nF, and $I_S$ = 10 nA to obtain $I_{OUT}$ of 1 μA. $N_{OPT}$ was determined by Equations (4) and (5) to be 16. Then, $C_{OPT}$ was calculated by $S/N_{OPT}$ to be 62.5 pF. $I_{OUT0}$ was calculated by Equations (1)–(3) to be 7.0 μA, and therefore the scaling factor λ was 1/7 μA = 0.14 μA. As a result, $I_{SFIN}$, $C_{FIN}$, and $S_{FIN}$ were 1.4 nA, 8.8 pF and 140 pF, respectively. Figure 7 shows the waveform of the output current with SPICE. The $I_{OUT}$ was measured to be 0.97 μA, which is 3% smaller than the target $I_{OUT}$. This result validates the proposed design method and flow.

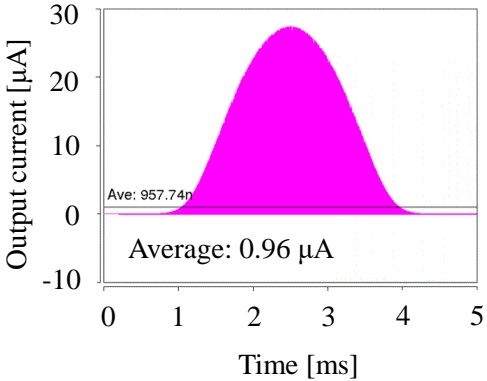

**Figure 7.** Output current waveform simulated by SPICE and the average value $I_{OUT}$.

## 5. Discussion

Electromagnetic VET has similar characteristics to magnetostrictive VET with respect to its relatively lower output open circuit voltage and low output impedance. Therefore, the design method proposed in this paper for clocked AC-DC charge pumps can also be applied to electromagnetic VET. The other types of VET—electrostatic, piezoelectric and triboelectric VET—require alternative types of power conversion than clocked AC-DC charge pump circuits.

SPICE simulation results showed that the clocked AC-DC charge pump circuit demonstrated in Section 4 has an input power, $P_{IN}^{CP}$, of 8.8 μW, resulting in a charge pump power efficiency, $\eta_{CP} \equiv P_{OUT}^{CP}/P_{IN}^{CP}$ of 23%. When a magnetostrictive VET with $V_{DD} = 0.5$ V and an output resistance of 500 Ω is used, the maximum available power $P_{OUT\_MAX}^{ET}$ is calculated as 31 μW. Therefore, overall power efficiency $\eta_{SYS} \equiv P_{OUT}^{CP}/P_{OUT\_MAX}^{ET}$ is estimated to be 8%.

In this paper, the impact of the parasitic capacitance of pumping capacitors ($C_P$) on (1) was disregarded for simplicity. Design Equations (4) and (5) will need to be extended for cases where $C_P$ cannot be ignored. When clocked AC-DC charge pump circuits are designed considering the output impedance of VET, another design flow, such as that described in [25], will be required in addition to the design flow proposed in Figure 6. Further study is required to outline an optimum design to maximize power efficiency.

## 6. Conclusions

In this paper, we proposed Equations (4) and (5) to determine the number of stages which enables designers to obtain clocked AC-DC charge pumps with the minimum circuit area based on the previously formulated output voltage and current equation. In addition, a design procedure was also presented to determine the capacitance of each pump capacitor and the size of each rectifying device when the amplitude of the voltage generated by the vibration energy transducer, target output voltage, and target output current are given. The design method and flow are demonstrated and validated by SPICE simulation with 1600 different sets of design parameters. When a clocked AC-DC charge pump was designed based on the optimum equations to have an output current of 1 μA at 2.5 V from a vibration energy harvester with an AC voltage amplitude of 0.5 V, the discrepancy in the output current was just 3% from that based on a standard design method. Clocked AC-DC charge pump circuits with the minimum circuit area can be designed based on the equations and design flow proposed in this paper.

**Author Contributions:** Conceptualization, T.T.; methodology, J.Y. and T.T.; software, J.Y.; validation, J.Y. and T.T.; formal analysis, J.Y. and T.T.; investigation, J.Y. and T.T.; writing—original draft preparation, T.T.; writing—review and editing, J.Y. and T.T.; funding acquisition, T.T. All authors have read and agreed to the published version of the manuscript.

**Funding:** This research was partially funded by the Micron Foundation.

**Acknowledgments:** This work was supported by VDEC, Synopsys, Inc., Cadence Design Systems, Inc. Rohm Corp. and the Micron Foundation. The authors wish to thank M. Futagawa, H. Hirano and S. Ota for technical discussion.

**Conflicts of Interest:** The authors declare no conflict of interest.

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
