# Peer review of "An Optimum Design of Clocked AC-DC Charge Pump Circuits for Vibration Energy Harvesting"

_electronics, doi:10.3390/electronics9122031_

Round 1

Reviewer 1 Report

Authors proposed an Optimum design of clocked ac-dc charge pump circuit for vibration energy harvesting. The topic is interested for scientific community. But, the novelty of the work is explained neither in the abstract nor in conclusions. English language is hardly readable in some points, and in general lacks clarity and grammar. The manuscript is not suited for publication in the present form. I suggest the authors to resubmit the manuscript after addressing the shortcomings listed below:

  • Conclusions should be more related to the research conducted. They should not be just a description of what has been done.
  • Line 30: define ‘VEH’
  • Line 39: define ‘EH’
  • line 43: you use just ET, because you explained this in line 21-22.
  • line 46: you use just VET, because you explained this in line 24.
  • Please revise the abbreviations in the manuscript.
  • Image quality of Figures 1 and 5 should be improved according to MDPI standard.
  • What is the efficiency of proposed clocked ac-dc charge pump circuits-based vibration energy harvester? Please compare with similar methods.
  • Line 65-66: authors mentioned that ‘none of them doesn’t discuss how the clocked AC-DC charge pumps should be designed.’ It doesn’t seem true. Authors should carefully study the literature. For example:

Kawauchi et al., A 2V 3.8 µW Fully-Integrated Clocked AC-DC Charge Pump with 0.5 V 500Ω Vibration Energy Harvester. In 2019 IEEE Asia Pacific Conference on Circuits and Systems (APCCAS) (pp. 329-332).

  • Are all the equations developed by the authors? If not then please cite appropriate references.
  • Please add discussion section before conclusion.

Reviewer 2 Report

In this paper, the author's design AC-DC charge pump circuits for vibration energy harvesting. There are a number of issues to be taken care of throughout the manuscript. Besides, the inclusion of some more information would strengthen the quality of the paper. 

Here are my concerns:

  1. [Abstract] should be more informative to present the findings of the work (including the results).
  2. The novelty of this work is unclear. In particular, the authors should clarify the novelty of this work.
  3. [Introduction] Should be more organized and informative. The last paragraph of the introduction must be modified and updated with the novelty, scientific soundness, technical issues, and methods used, as well as the quantified results of the proposed work. The current content does not have any adequate information. It is better to outline the structure of the paper at the end of Section 1.
  4. What happened if the proposed circuit is added with the electromagnetic and triboelectric harvesters?
  5. The figure quality needs to be improved in this manuscript.
  6. The authors should include the comparison results with previously reported works.
  7. [Conclusion] should summarize the key idea and theory.

Reviewer 3 Report

Introduction chapter. The novelty of the work is not clarified and literature review does not support this goal. Authors should provide a deep review of the state of the art.

The introduction should be restyled to better highlight main novelties introduced by this paper and advantages. Authors should highlight the novelty and compare it with novelty of other authors’ works.

In the article is duplicated material from line 41 to line 63.

Organization of the manuscript should be improved. It is necessary to clearly describe the structure of the article in the introduction.

Results. Where is the comparison with existing literature? In addition, where is the critical analysis of results? 

The article is prepared in a very poor quality, the images are informative and of poor quality, the conclusions are unacceptable.

In my opinion, the article is not appropriate for this journal.

Round 2

Reviewer 1 Report

Dear Authors, the revised manuscript is significantly improved. But, the writing style of the manuscript is needed more improvement. Some sentences are difficult to read and understand.

Reviewer 2 Report

In the revised manuscript, the authors could not address all of my comments with proper explanation. However, the following comments have not been well addressed yet, which are important for understanding and improving the work. The overall presentation of this work must be improved. The comments must be well addressed before considering the publication of this manuscript.

  1. I didn’t find any suitable description of the novelty this work.

“Novelty of this work is unclear. In particular, the authors should clarify the novelty this work.”

  1. [Discussion] This part is a very poor presentation. The authors should discuss more details about the proposed works.
  2. How the author’s optimized the design of the clocked AC-DC charge pump? Please add a more detailed description of the optimized parameters.
  3. I didn’t find any results about the efficiency of the circuit. The authors should discuss efficiency and include efficiency results.
  4. “Two different definitions for power efficiency were used for comparison”. I didn’t find any results of power efficiency. Figure 5 (a), what is the population? Please explain…

Reviewer 3 Report

Accept. The article has been heavily corrected.

Round 3

Reviewer 1 Report

The manuscript is improved significantly. The manuscript can be accepted in the present form.

Reviewer 2 Report

The paper quality is improved after the revision. Now, I recommend the manuscript to be published in its current form.